# Mobility-Embedded POIs: Learning What A Place Is and How It Is Used from Human Movement

**Maria Despoina Siampou** [1 2] **Shushman Choudhury** [2] **Shang-Ling Hsu** [1] **Neha Arora** [2] **Cyrus Shahabi** [1]

## Abstract

Recent progress in geospatial foundation models highlights the importance of learning general-purpose representations for real-world locations, particularly points-of-interest (POIs) where human activity concentrates. Existing approaches, however, focus primarily on place *identity* derived from static textual metadata, or learn representations tied to trajectory context, which capture movement regularities rather than how places are actually used (i.e., POI's *function*). We argue that POI function is a missing but essential signal for general POI representations. We introduce **Mobility-Embedded POIs (ME-POIs)**, a framework that augments POI embeddings derived, from language models with large-scale human mobility data to learn POI-centric, context-independent representations grounded in real-world usage. ME-POIs encodes individual visits as temporally contextualized embeddings and aligns them with learnable POI representations via contrastive learning to capture usage patterns across users and time. To address long-tail sparsity, we propose a novel mechanism that propagates temporal visit patterns from nearby, frequently visited POIs across multiple spatial scales. We evaluate ME-POIs on five newly proposed map enrichment tasks, testing its ability to capture both the identity and function of POIs. Across all tasks, augmenting text-based embeddings with ME-POIs consistently outperforms both text-only and mobility-only baselines. Notably, ME-POIs trained on mobility data alone can surpass text-only models on certain tasks, highlighting that POI function is a critical component of accurate and generalizable POI representations.

[1]Department of Computer Science, University of Southern California, Los Angeles, CA, USA [2]Google Research, Mountain View, CA, USA. Correspondence to: Maria Despoina Siampou <siampou@google.com, siampou@usc.edu>.

*Proceedings of the 43$^{rd}$ International Conference on Machine Learning*, Seoul, South Korea. PMLR 306, 2026. Copyright 2026 by the author(s).

## 1. Introduction

The increasing availability of large-scale geospatial data, together with advances in machine learning, is reshaping our ability to model and reason about urban environments (Lee & Kang, 2015; Mai et al., 2024). In such environments, points-of-interest (POIs), i.e., places that people visit during their everyday life such as restaurants, metro stations, and convenience stores, serve as core units of urban structure and activity. Consequently, learning representations that capture the intrinsic semantics of a place, including both its identity *(what a place is)* and its function *(how a place is used)*, is fundamental to a range of geospatial applications, including digital map maintenance, location recommendation, and urban analytics (Siampou et al., 2025a).

To capture the *identity* of POIs, existing approaches primarily focus on encoding the static attributes of places (Li et al., 2022; 2023; Cheng et al., 2025). In particular, recent methods leverage large language models (LLMs) to learn POI representations, due to their ability to encode extensive geographic and semantic knowledge from massive internet-scale data (Manvi et al., 2024). These approaches have demonstrated that with carefully designed prompts, often augmented with map data (e.g., geo-coordinates, POI category) and contextual neighborhood information (e.g., categories of closest POIs), one can effectively unlock the vast latent geospatial knowledge embedded within these models. However, this exclusive reliance on static signals can limit performance in dynamic urban environments, where metadata can be missing (e.g., new POIs) or outdated. Furthermore, similar text does not always imply similar function. For instance, two coffee shops on the same street can have the same static attributes and neighborhood context, but serve very different roles in practice: one may be a chain characterized by quick, high-turnover visits, while the other may be a local café where customers work and socialize and spend more time per visit. By ignoring the dynamic behavioral signals that distinguish places, static models can conflate functionally distinct locations.

Meanwhile, prior work on mobility data has primarily used sequence-based models to predict the next visited location from surrounding trajectory context (Feng et al., 2017; Wan et al., 2021; Lin et al., 2021; Xue et al., 2021; Hsu et al.,

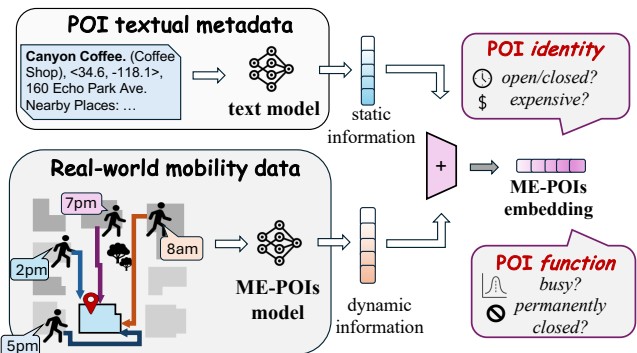

*Figure 1.* **Illustration of ME-POIs.** ME-POIs augment static text-based POI representations with mobility-derived signals, to learn POI embeddings that capture their identity and function.

2024). While effective at modeling movement regularities, these learned POI embeddings are optimized for trajectory prediction and therefore do not capture POI *function*. For instance, a gym and a bar near the same office may both be frequently visited after work; sequence-based models encode their embeddings to reflect this shared post-work usage, capturing similar movement patterns while missing intrinsic differences such as operating hours or the type of activity offered. As a result, the these representations are inherently context-dependent, reflecting how a place appears within specific sequences rather than providing a universal, context-independent encoding of POI function.

In this work, we argue that POI function is a missing but essential signal for general POI representations. We address this by introducing **Mobility-Embedded POIs (ME-POIs)**, a framework that augments static POI embeddings derived from text models with large-scale human mobility, producing representations that capture the **intrinsic semantics** of each place, which we define as encoding both the POI's identity *(what a place is)* and its function *(how it is used)*. We present the main idea of the framework in Figure 1. Starting from visit sequences, our approach encodes each visit as a contextualized embedding that reflects the static attributes of the POI and its temporal context within mobility patterns. These visit-level embeddings are then aligned with a learnable POI embedding via contrastive learning, ensuring that each POI representation incorporates aggregated behavioral information over time and across users. To address the common challenge of data sparsity for rarely visited POIs, we propose a distribution transfer mechanism that propagates temporal usage patterns from close by, frequently visited POIs, across multiple spatial scales, to those with limited data. This multi-scale strategy captures both local and regional behavioral trends and yields high-quality POI embeddings even in the long tail of the visit distribution.

We evaluate ME-POIs on two real-world large-scale mobility datasets across five newly proposed tasks critical for automated map enrichment and maintenance: weekly open-

ing hours prediction, permanent closure detection, visit intent classification, weekly busyness estimation, and price level classification. These tasks are strategically chosen to assess both intrinsic semantics, evaluating the model's ability to capture static identity (e.g., price, opening hours) as well as dynamic functional states (e.g., visit intent, busyness, closure status). Moreover, these attributes are often incomplete, outdated, or difficult to maintain at scale, underscoring the practical value of our mobility-informed POI representations. Across all benchmarks, augmenting strong text-based embeddings, including those from OpenAI and Gemini models, with ME-POIs yields consistent and substantial improvements, with gains of up to $16.2\%$ for opening hours, $81.9\%$ for visit intent, $75.1\%$ for price level, and $6.5\%$ in F1 for permanent closure detection, as well as up to a $24.7\%$ reduction in MAE for busyness estimation. These results demonstrate that single POI embeddings learned by ME-POIs can support a diverse set of downstream tasks, highlighting the versatility of our framework for enriching POI representations. Notably, ME-POIs outperforms all mobility-based baselines both with and without explicit textual POI semantics, with the latter even surpassing some text-based embeddings on certain tasks (e.g., GEMINI embeddings for price level classification), further emphasizing the strength of our approach. In sum, our contributions are:

- We propose **Mobility-Embedded POIs (ME-POIs)**, a framework that augments static, text-based POI embeddings with mobility-derived representations.
- We introduce a new mobility-based objective to learn POI-centric embeddings that encode POI *identity* and *function* from visit sequences, rather than local trajectory transitions.
- We propose a contrastive learning paradigm that aligns visit-level embeddings with learnable POI embeddings, and a novel multi-scale visit distribution transfer mechanism to address sparsity in long-tail, rarely visited POIs.
- We evaluate ME-POIs on a set of map enrichment tasks, demonstrating consistent improvements over both text- and mobility-based baselines.

## 2. Problem Formulation

Let $\mathcal{P} = \{p_1, \ldots, p_N\}$ denote a set of POIs within a geographic region. Each POI $p \in \mathcal{P}$ is associated with a location $x_p \in \mathbb{R}^2$ and textual metadata (e.g., description), from which we obtain a static embedding $z_p^{\text{static}} \in \mathbb{R}^d$ using a pretrained text embedding model. Let, also, $\mathcal{S} = \{s_1, \ldots, s_K\}$ denote a collection of user visit sequences, where each sequence $s_k = (v_1, \ldots, v_{L_k})$ is temporally ordered. Each visit is defined as $v_i = (p_i, t_i^a, t_i^d)$, where $p_i \in \mathcal{P}$ is the visited POI and $t_i^a, t_i^d \in \mathbb{R}$ are arrival and departure times.

**Objective.** Given static POI embeddings $\{z_p^{\text{static}}\}_{p \in \mathcal{P}}$ and mobility data $\mathcal{S}$, our goal is to learn a POI-centric representation $z_p^{\text{ME}} \in \mathbb{R}^d$ for each $p \in \mathcal{P}$ that integrates textual

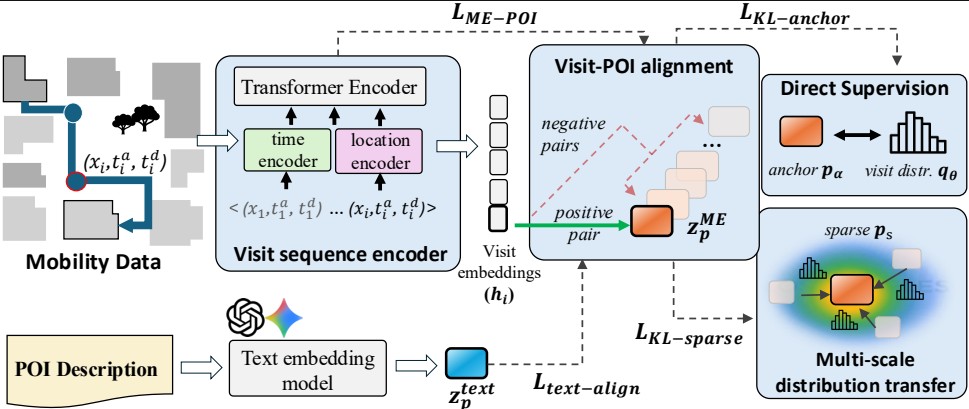

*Figure 2.* **Overview of ME-POIs pretraining.** The framework includes: (i) a transformer-based visit sequence encoder, (ii) contrastive alignment of contextualized visits ($h$) with global POI embeddings ($z_p^{\text{ME}}$) to capture usage patterns, (iii) multi-scale distribution transfer to propagate temporal visit information to under-visited POIs ($p_s$), (iv) direct supervision on anchor POIs ($p_a$) to regularize embeddings via visit distribution prediction, and (v) an auxiliary text-alignment objective to ground POI embeddings ($z_p^{\text{ME}}$) in textual semantics ($z_p^{\text{text}}$).

semantics with longitudinal visitation patterns. Formally, we aim to learn a function $f$ such that

$$z_p^{\text{ME}} = f\left(z_p^{\text{static}}, \mathcal{S}_p\right), \tag{1}$$

where $\mathcal{S}_p \subseteq \mathcal{S}$ denotes the subset of visit sequences containing visits to $p$. The resulting embeddings encode the POIs intrinsic semantics, including their *identity* and *function*.

## 3. Methodology

In this section, we present the components our ME-POIs framework, as illustrated in Figure 2.

### 3.1. Visit Sequence Encoder

**Visit Feature Encoding.** Given a visit sequence $s \in \mathcal{S}$, each visit $v_i \in s$ is characterized by three attributes describing a user's interaction with a POI $p_i$: the geographical coordinates $x_{p_i} \in \mathbb{R}^2$, and the arrival and departure times $t_i^a, t_i^d \in \mathbb{R}$. We transform each attribute into a fixed-dimensional vector using three factorized encoders. Specifically, we utilize the multiscale location encoder from Space2Vec (Mai et al., 2020), which we denote as $\lambda_\theta : \mathbb{R}^2 \to \mathbb{R}^{d_l}$, to capture spatial relationships at multiple scales from local neighborhoods to broader regional context[1]. We further use Time2Vec (Kazemi et al., 2019) to encode arrival and departure times separately, denoted as $g_\eta, g_\zeta : \mathbb{R} \to \mathbb{R}^{d_t}$, which helps us capture the distinct temporal patterns in visit start times and durations.

To preserve the distinct contributions of each attribute in the visit representation, we explicitly concatenate their encodings to form the initial vector for $v_i$:

$$\tilde{h}_i^{(0)} = [\lambda_\theta(x_i) \,\|\, g_\eta(t_i^a) \,\|\, g_\zeta(t_i^d)] \in \mathbb{R}^{d_h}, \tag{2}$$

[1]More advanced location encoders, such as Poly2Vec (Siampou et al., 2025b), could be used when POIs are represented as richer spatial geometries (e.g., building footprints as polygons).

where $d_h = d_l + 2d_t$ and $[\cdot \,\|\, \cdot]$ denotes vector concatenation.

**Sequence Modeling.** After encoding each visit into a fixed-dimensional vector, we aim to contextualize these representations to capture dependencies and patterns within the visit sequence, which are essential for understanding the functional usage of POIs. For this, we apply a multi-layer Transformer encoder model (Vaswani et al., 2017), which is a standard choice for capturing temporal and co-visitation patterns in trajectory modeling (Xue et al., 2021; Hsu et al., 2024). To preserve the temporal order of visits within a sequence of visit embeddings, $\tilde{H}^{(0)} = (\tilde{h}_1^{(0)}, \dots, \tilde{h}_L^{(0)})$, we first augment the sequence with a fixed sinusoidal positional encoding $\text{PE}(i)$ as follows:

$$h_i^{(0)} = \tilde{h}_i^{(0)} + \text{PE}(i) \tag{3}$$

The position-aware embeddings $H^{(0)} = (h_1^{(0)}, \dots, h_L^{(0)})$ are then fed into the Transformer. Each layer consists of multi-head self-attention followed by a feedforward network (FFN) with residual connections and layer normalization:

$$H' = \text{LayerNorm}(H^{(0)} + \text{MultiHead}(H^{(0)})) \tag{4}$$

$$H^{(1)} = \text{LayerNorm}(H' + \text{FFN}(H')) \tag{5}$$

Multi-head attention is computed as:

$$\text{MultiHead}(H) = [\text{head}_1 \| \cdots \| \text{head}_j] W^O, \tag{6}$$

$$\text{head}_i = \text{Softmax}\left(\frac{H W_i^Q (H W_i^K)^\top}{\sqrt{d_k}}\right) H W_i^V, \tag{7}$$

where $W_i^Q, W_i^K, W_i^V \in \mathbb{R}^{d_h \times d_k}$ and $W^O \in \mathbb{R}^{j d_k \times d_h}$ are learnable parameters. After $N$ stacked layers, the Transformer produces the final contextualized visit embeddings:

$$H = (h_1, \dots, h_L), \quad h_i \in \mathbb{R}^{d_h} \tag{8}$$

## 3.2. Global POI Alignment via Contrastive Learning

While the sequence encoder captures transient movement dynamics, our ultimate goal is to learn global, context-independent embeddings for each POI. To do so, we define a learnable embedding matrix $\mathbf{Z}^{\mathrm{ME}} \in \mathbb{R}^{|\mathcal{P}| \times d_h}$, where each vector $z_p^{\mathrm{ME}}$ serves as the global prototype for POI $p$. Formally, given a contextualized visit embedding $h_i$ corresponding to a visit to POI $p$, we aim to align $h_i$ with its prototype $z_p^{\mathrm{ME}}$, while distinguishing it from the prototypes of other POIs. We achieve this by minimizing the InfoNCE loss (Oord et al., 2018; Radford et al., 2021), treating $(h_i, z_p^{\mathrm{ME}})$ as a positive pair and the prototypes of other POIs in the current minibatch as negatives:

$$\mathcal{L}_{\mathrm{ME\text{-}POI}}(h_i, z_p^{\mathrm{ME}}) = -\log \frac{\exp(\mathrm{sim}(h_i, z_p^{\mathrm{ME}})/\tau)}{\sum\limits_{p' \in \mathcal{P}_{\mathrm{batch}}} \exp(\mathrm{sim}(h_i, z_{p'}^{\mathrm{ME}})/\tau)},$$
(9)

where $\mathrm{sim}(a, b) = \frac{a^\top b}{\|a\|\|b\|}$ denotes cosine similarity and $\tau$ is a temperature hyperparameter.

Intuitively, this alignment encourages the global prototype to act as a functional centroid, aggregating usage patterns across diverse visits while suppressing the noise inherent in individual user schedules. This process naturally captures both the POI's *function*, derived from consistent temporal behaviors (e.g., dwell times, daily cycles), and its unique *identity*, as the contrastive objective forces the representation to be distinct from even spatially proximate neighbors.

## 3.3. Multi-Scale Distribution Transfer for Sparse POIs

A common challenge in modeling human mobility is the long-tail distribution of visit frequencies. In practice, only a small fraction of POIs is frequently observed in the data, whereas most locations record insufficient visits (Chen et al., 2021; Xu et al., 2024). This data imbalance limits the effectiveness of our contrastive learning module, since the global prototypes of sparsely visited POIs are updated from only a handful of visits. As a result, their embeddings may fail to reliably reflect their underlying functional semantics.

To address this issue, we introduce a multi-scale visit distribution transfer mechanism that injects structured temporal priors into sparse POI embeddings by leveraging visitation patterns from nearby, data-rich locations. This choice is motivated by our observation that human mobility is strongly guided by spatial context: POIs of the same urban environment tend to exhibit similar activity patterns (e.g., similar peak hours), driven by shared land use, accessibility, commuting flows, and surrounding population dynamics. Thus, transferring knowledge from frequent POIs to their sparse neighbors can stabilize the latter's embeddings.

Formally, we partition the POIs into a set of anchor POIs, $\mathcal{P}_{\mathrm{anchor}}$, consisting of the top-$k$ POIs with the highest total visit counts, and a set of sparse POIs, denoted $\mathcal{P}_{\mathrm{sparse}}$. For each anchor POI $p_a \in \mathcal{P}_{\mathrm{anchor}}$, we construct an empirical visit distribution $r_{p_a} \in \Delta^T$ by aggregating visits into $T$ fixed temporal bins (e.g., hourly slots over a week) and normalizing the resulting histogram. These distributions serve as stable temporal priors for the transfer process.

However, simply transferring priors from the nearest anchor is insufficient, as urban dynamics exist at multiple resolutions. Fine-grained spatial proximity captures localized effects, such as neighboring establishments sharing similar peak hours, while broader scales reflect neighborhood and district-level patterns (e.g., a commercial area). To capture these hierarchical dependencies, we employ a multi-scale kernel mechanism parameterized by $M$ bandwidths $\{\sigma_m\}_{m=1}^M$. For a sparse POI $p_s \in \mathcal{P}_{\mathrm{sparse}}$, the spatial influence weight of an anchor $p_a$ at scale $m$ is computed using a normalized Gaussian kernel:

$$\alpha_{p_s, p_a}^{(m)} = \frac{\exp\left(-\frac{\|x_{p_s} - x_{p_a}\|^2}{2\sigma_m^2}\right)}{\sum_{p_a' \in \mathcal{P}_{\mathrm{anchor}}} \exp\left(-\frac{\|x_{p_s} - x_{p_a'}\|^2}{2\sigma_m^2}\right)},$$
(10)

where $x_{p_s}$ and $x_{p_a}$ denote the coordinates of the sparse POI and anchor, respectively.

We then estimate the expected temporal activity for each sparse POI $p_s \in \mathcal{P}_{\mathrm{sparse}}$ by aggregating the empirical distributions of anchor POIs across multiple spatial scales:

$$\tilde{r}_{p_s} = \frac{1}{M} \sum_{m=1}^M \sum_{p_a \in \mathcal{P}_{\mathrm{anchor}}} \alpha_{p_s, p_a}^{(m)} \cdot r_{p_a},$$
(11)

To inject this temporal prior into the embedding space, we require the learned prototype $z_{p_s}^{\mathrm{ME}}$ to predict a visitation distribution. For this, we map $z_{p_s}^{\mathrm{ME}}$ through a multi-layer perceptron followed by a softmax function:

$$q_\theta(p_s) = \mathrm{softmax}(\mathrm{MLP}(z_{p_s}^{\mathrm{ME}})),$$
(12)

where $\mathrm{MLP}(\cdot)$ denotes a neural network with one hidden layer and ReLU activation.

Finally, we train the model to align the predicted distribution $q_\theta(p_s)$ with the transferred prior $\tilde{r}_{p_s}$ using an auxiliary KL divergence loss:

$$\mathcal{L}_{\mathrm{KL\text{-}sparse}} = \sum_{p_s \in \mathcal{P}_{\mathrm{sparse}}} \mathrm{KL}\left(\tilde{r}_{p_s} \| q_\theta(p_s)\right)$$
(13)

## 3.4. Direct Supervision for Anchor POIs

We also directly supervise the embeddings of anchor POIs to ensure that their global prototypes faithfully encode empirically observed temporal usage patterns throughout training.

Following the same prediction mechanism used for sparse POIs in Section 3.3, we map each prototype $z_{p_a}^{\text{ME}}$ of an anchor POI $p_a \in \mathcal{P}_{\text{anchor}}$ to a visitation distribution:

$$q_\theta(p_a) = \text{softmax}(\text{MLP}(z_{p_a}^{\text{ME}})), \quad (14)$$

where $\text{MLP}(\cdot)$ denotes the same network used for sparse POIs. We then minimize the KL divergence between the empirical and predicted distributions:

$$\mathcal{L}_{\text{KL-anchor}} = \sum_{p_a \in \mathcal{P}_{\text{anchor}}} \text{KL}\left(r_{p_a} \,\|\, q_\theta(p_a)\right) \quad (15)$$

### 3.5. Alignment with Text Embeddings

Our learned mobility embeddings are designed to enrich static POI text embeddings. To extract rich semantic and spatial information from text, we follow the prompt design methodology introduced in GeoLLM (Manvi et al., 2024), which demonstrates how to construct effective LLM prompts to extract geospatial knowledge. Specifically, we describe each POI using both its intrinsic attributes (i.e., coordinates, category, address) and local neighborhood context (i.e., the direction and distance of nearby POIs). This approach ensures that the resulting text embeddings encode meaningful geospatial and contextual information, which we then align with our mobility embeddings. Examples of the constructed prompts are provided in Appendix A.2.

To encourage the ME-POIs embeddings $z_p^{\text{ME}} \in \mathbb{R}^{d_h}$ to capture complementary semantic information, we project the text embeddings $z_p^{\text{static}} \in \mathbb{R}^{d_u}$ into the mobility embedding space via a linear mapping $W \in \mathbb{R}^{d_h \times d_u}$. We then maximize the cosine similarity between the mobility and projected text embeddings. This objective encourages $z_p^{\text{ME}}$ to incorporate semantic signals from textual descriptions while preserving information derived from mobility patterns:

$$\mathcal{L}_{\text{text-align}} = \sum_{p \in \mathcal{P}} \left[1 - \cos\left(z_p^{\text{ME}}, W z_p^{\text{static}}\right)\right], \quad (16)$$

where $\cos(\cdot, \cdot)$ denotes cosine similarity.

### 3.6. Model Optimization

**Pretraining.** We pretrain the model by jointly optimizing a primary contrastive objective, $\mathcal{L}_{\text{ME-POI}}$, together with three auxiliary losses that (i) regularize anchor POIs, (ii) transfer temporal patterns to sparse POIs, and (iii) align mobility embeddings with text semantics:

$$\mathcal{L} = \mathcal{L}_{\text{ME-POI}} + \lambda_a \, \mathcal{L}_{\text{KL-anchor}} + \lambda_s \, \mathcal{L}_{\text{KL-sparse}} + \lambda_t \, \mathcal{L}_{\text{text-align}}, \quad (17)$$

where $\lambda_a$, $\lambda_s$, and $\lambda_t$ weight the auxiliary terms.

**Fine-Tuning.** For downstream tasks, we freeze the pretrained embeddings and train lightweight task-specific heads.

For each POI $p$, the mobility embedding $z_p^{\text{ME}}$ and text embedding $z_p^{\text{static}}$ are independently projected via small MLPs, concatenated, and passed to a task-specific prediction head:

$$\hat{y}_p = \text{MLP}_{\text{head}}\left([\text{MLP}_p(z_p^{\text{ME}}) \,\|\, \text{MLP}_t(z_p^{\text{static}})]\right) \quad (18)$$

All MLPs consist of one hidden layer with ReLU activation.

## 4. Experiments

### 4.1. Experimental Setup

**Datasets.** We use two large-scale, anonymized human mobility datasets from Veraset[2], covering Los Angeles County and Houston. The first spans one year, while the second covers 20 days. More details are provided in Appendix A.1.1.

**Baselines.** We compare ME-POIs against state-of-the-art text and mobility-based baselines. For text embeddings, we use MPNET (Song et al., 2020), E5 (Wang et al., 2022), GTR-T5 (Ni et al., 2022), NOMIC (Nussbaum et al., 2024), OPENAI, and GEMINI. For mobility-based, we consider SKIP-GRAM (Mikolov et al., 2013), POI2VEC (Feng et al., 2017), GEO-TEASER (Zhao et al., 2017), TALE (Wan et al., 2021), HIER (Shimizu et al., 2020), CTLE (Lin et al., 2021), DEEPMOVE (Feng et al., 2018), STAN (Luo et al., 2021), GRAPH-FLASHBACK (Rao et al., 2022), GET-NEXT (Yang et al., 2022), and TRAJGPT (Hsu et al., 2024). All models are evaluated via frozen-embedding probing.

**Downstream Tasks.** We evaluate our approach on five map enrichment tasks: (i) multi-label classification of weekly **open hours**, (ii) binary classification of **permanent closure** status, (iii) **visit intent** classification, derived from aggregated navigation queries and discretized into four classes from least to most popular, (iv) prediction of **busyness**, as a weekly average of hourly activity levels, and (v) **price level** classification. Ground-truth labels for opening hours and permanent closures are obtained from SafeGraph[3], while the remaining are sourced from Google Maps. Permanent closure is evaluated only on Los Angeles due to limited label quality in Houston. For each task, we report two standard evaluation metrics appropriate to the prediction objective. Additional task details are provided in Appendix A.1.2.

### 4.2. Main Results

**Augmentation of text-based models.** Tables 1 and 2 report the performance of text embedding models with and without ME-POIs on the Los Angeles and Houston datasets. Across both datasets and all tasks, adding ME-POIs consistently improves performance of text models, often by a substantial margin. The improvements are particularly evident for dynamic, function-focused tasks, with up to 81.9%

---

[2] https://www.veraset.com
[3] https://www.safegraph.com/

*Table 1.* Performance of text-based baselines in Los Angeles. Results report the mean over 5 runs. Relative improvements from adding ME-POIs are highlighted next to each metric.

| Method | Open Hours (↑) F1 / (↑) AUROC | Permanent Closure (↑) F1 / (↑) AUPRC | Visit Intent (↑) F1 / (↑) AUPRC | Busyness (↓) MAE / (↑) Cosine | Price Level (↑) Accuracy / (↑) F1 |
|---|---|---|---|---|---|
| ME-POIs (w/o $\mathcal{L}_{\text{text-align}}$) | 0.540 / 0.703 | 0.757 / 0.154 | 0.263 / 0.337 | 0.159 / 0.878 | 0.600 / 0.308 |
| MPNET | 0.542 / 0.726 | 0.736 / 0.172 | 0.270 / 0.382 | 0.171 / 0.873 | 0.615 / 0.306 |
| MPNET + ME-POIs | 0.628(↑15.8%) / 0.783(↑7.8%) | 0.766(↑4.1%) / 0.181(↑5.2%) | 0.352(↑30.3%) / 0.410(↑7.3%) | 0.138(↓19.2%) / 0.896(↑2.6%) | 0.662(↑7.6%) / 0.337(↑10.1%) |
| E5 | 0.540 / 0.722 | 0.738 / 0.176 | 0.184 / 0.344 | 0.169 / 0.872 | 0.521 / 0.189 |
| E5 + ME-POIs | 0.601(↑11.3%) / 0.751(↑4.0%) | 0.786(↑6.5%) / 0.185(↑5.1%) | 0.330(↑79.4%) / 0.391(↑13.6%) | 0.142(↓15.9%) / 0.892(↑2.2%) | 0.632(↑21.3%) / 0.322(↑70.4%) |
| GTR-T5 | 0.547 / 0.721 | 0.767 / 0.173 | 0.241 / 0.365 | 0.168 / 0.873 | 0.586 / 0.278 |
| GTR-T5 + ME-POIs | 0.618(↑12.9%) / 0.767(↑6.4%) | 0.774(↑0.9%) / 0.178(↑2.9%) | 0.332(↑37.8%) / 0.398(↑9.0%) | 0.141(↓16.0%) / 0.894(↑2.4%) | 0.654(↑11.6%) / 0.334(↑20.1%) |
| NOMIC | 0.539 / 0.723 | 0.749 / 0.173 | 0.230 / 0.361 | 0.168 / 0.873 | 0.614 / 0.297 |
| NOMIC + ME-POIs | 0.619(↑14.8%) / 0.771(↑6.6%) | 0.762(↑1.7%) / 0.182(↑5.2%) | 0.332(↑44.4%) / 0.403(↑11.6%) | 0.143(↓14.8%) / 0.894(↑2.4%) | 0.659(↑7.3%) / 0.336(↑13.1%) |
| OPENAI-SMALL | 0.547 / 0.732 | 0.695 / 0.184 | 0.260 / 0.390 | 0.167 / 0.874 | 0.637 / 0.320 |
| OPENAI-SMALL + ME-POIs | 0.632(↑15.5%) / 0.780(↑6.6%) | 0.696(↑0.1%) / 0.186(↑1.2%) | 0.353(↑35.8%) / 0.414(↑6.1%) | 0.138(↓17.3%) / 0.896(↑2.5%) | 0.675(↑4.3%) / 0.345(↑7.8%) |
| OPENAI-LARGE | 0.548 / 0.738 | 0.750 / 0.181 | 0.271 / 0.404 | 0.169 / 0.873 | 0.654 / 0.329 |
| OPENAI-LARGE + ME-POIs | 0.637(↑16.2%) / 0.783(↑6.1%) | 0.770(↑2.7%) / 0.185(↑2.2%) | 0.368(↑35.8%) / 0.435(↑7.6%) | 0.136(↓19.5%) / 0.897(↑2.7%) | 0.684(↑4.6%) / 0.350(↑6.4%) |
| GEMINI | 0.548 / 0.716 | 0.756 / 0.181 | 0.199 / 0.367 | 0.190 / 0.856 | 0.559 / 0.234 |
| GEMINI + ME-POIs | 0.613(↑11.9%) / 0.761(↑6.3%) | 0.753(↓0.4%) / 0.185(↑2.2%) | 0.362(↑81.9%) / 0.423(↑15.2%) | 0.143(↓24.7%) / 0.894(↑4.4%) | 0.672(↑20.2%) / 0.345(↑47.4%) |

*Table 2.* Performance of text-based baselines in Houston. Results report the mean over 5 runs. Relative improvements from adding ME-POIs are highlighted next to each metric.

| Method | Open Hours (↑) F1 / (↑) AUROC | Visit Intent (↑) F1 / (↑) AUPRC | Busyness (↓) MAE / (↑) Cosine | Price Level (↑) Accuracy / (↑) F1 |
|---|---|---|---|---|
| ME-POIs (w/o $\mathcal{L}_{\text{text-align}}$) | 0.519 / 0.604 | 0.270 / 0.314 | 0.182 / 0.867 | 0.564 / 0.276 |
| MPNET | 0.653 / 0.739 | 0.331 / 0.416 | 0.164 / 0.886 | 0.599 / 0.248 |
| MPNET + ME-POIs | 0.725(↑11.0%) / 0.803(↑8.6%) | 0.374(↑12.9%) / 0.440(↑5.7%) | 0.137(↓16.4%) / 0.903(↑1.9%) | 0.687(↑14.6%) / 0.344(↑38.7%) |
| E5 | 0.640 / 0.754 | 0.229 / 0.389 | 0.163 / 0.886 | 0.549 / 0.177 |
| E5 + ME-POIs | 0.690(↑7.8%) / 0.780(↑3.4%) | 0.368(↑60.7%) / 0.412(↑5.9%) | 0.143(↓12.2%) / 0.901(↑1.6%) | 0.635(↑15.6%) / 0.300(↑69.4%) |
| GTR-T5 | 0.624 / 0.742 | 0.257 / 0.397 | 0.162 / 0.887 | 0.549 / 0.177 |
| GTR-T5 + ME-POIs | 0.713(↑14.2%) / 0.782(↑3.7%) | 0.370(↑61.5%) / 0.419(↑5.5%) | 0.141(↓12.9%) / 0.902(↑1.6%) | 0.645(↑17.4%) / 0.310(↑75.1%) |
| NOMIC | 0.721 / 0.806 | 0.268 / 0.383 | 0.162 / 0.887 | 0.578 / 0.212 |
| NOMIC + ME-POIs | 0.738(↑2.3%) / 0.813(↑0.8%) | 0.366(↑36.5%) / 0.410(↑7.0%) | 0.143(↓11.7%) / 0.901(↑1.5%) | 0.667(↑15.4%) / 0.326(↑53.7%) |
| OPENAI-SMALL | 0.654 / 0.761 | 0.314 / 0.424 | 0.161 / 0.887 | 0.595 / 0.233 |
| OPENAI-SMALL + ME-POIs | 0.743(↑13.6%) / 0.805(↑5.7%) | 0.398(↑26.7%) / 0.454(↑7.0%) | 0.137(↓14.9%) / 0.904(↑1.9%) | 0.729(↑22.5%) / 0.367(↑57.5%) |
| OPENAI-LARGE | 0.702 / 0.788 | 0.345 / 0.443 | 0.162 / 0.888 | 0.601 / 0.244 |
| OPENAI-LARGE + ME-POIs | 0.761(↑8.40%) / 0.824(↑4.57%) | 0.412(↑19.42%) / 0.475(↑7.2%) | 0.136(↓16.0%) / 0.906(↑2.0%) | 0.758(↑26.12%) / 0.383(↑56.97%) |
| GEMINI | 0.676 / 0.756 | 0.268 / 0.419 | 0.185 / 0.866 | 0.549 / 0.177 |
| GEMINI + ME-POIs | 0.741(↑9.62%) / 0.801(↑5.95%) | 0.392(↑46.27%) / 0.445(↑6.2%) | 0.142(↓23.2%) / 0.901(↑4.0%) | 0.634(↑15.48%) / 0.304(↑71.75%) |

and 6.5% increases in F1 for visit intent and permanent closure, respectively, and a 24.7% reduction in MAE for busyness. This aligns with our motivation: while strong text embedding baselines can capture the descriptive attributes of a place and often infer some coarse behavioral signals from web sources, they do not encode how places are actually used over time by people in their everyday activities. Interestingly, the identity-focused tasks of weekly opening hours and price level also show notable improvements, with F1 increasing up to 15.8% for opening hours and 75.1% for price level classification. While text embeddings capture attribute-related information, this knowledge is often incomplete, and biased toward popular POIs that are well-documented. By incorporating mobility, ME-POIs refine these representations, anchoring them to real-world visitation patterns over time and across users, which helps disambiguate POIs and fill gaps in textual metadata. Notably, even ME-POIs without any text information (the ME-POIs w/o $\mathcal{L}_{\text{text-align}}$ variant) outperforms strong text-only models in some cases, such as GEMINI on price level classification,

highlighting the rich signal contained in real-world mobility. Overall, these findings emphasize that encoding the intrinsic semantics of POIs (including both their *identity* and *function*) is essential for effective POI representations, and that integrating mobility with text embeddings produces more informative and generalizable POI embeddings.

**Comparison to mobility-based baselines.** We evaluate state-of-the-art mobility baselines against ME-POIs on the Los Angeles and Houston datasets, as reported in Tables 3 and 4. ME-POIs consistently outperform all baselines across both dynamic and static tasks. Notably, the second-best performance is achieved by ME-POIs trained without text alignment (the ME-POIs w/o $\mathcal{L}_{\text{text-align}}$ variant), indicating that the improvements arise primarily from our architecture rather than textual metadata. Since mobility-based models focus on capturing user movement patterns for next-location prediction, they fail to encode POI *identity* and *function*, limiting their effectiveness on our tasks. In contrast, ME-POIs explicitly aggregates information across visits through its contrastive alignment module, directly

*Table 3.* Comparison to mobility-based models in Los Angeles. Results averaged over 5 runs. **Best** and second best values are highlighted.

| Method | Open Hours ($\uparrow$) F1 / ($\uparrow$) AUROC | Permanent Closure ($\uparrow$) F1 / ($\uparrow$) AUPRC | Visit Intent ($\uparrow$) F1 / ($\uparrow$) AUPRC | Busyness ($\downarrow$) MAE / ($\uparrow$) Cosine | Price Level ($\uparrow$) Accuracy / ($\uparrow$) F1 |
|---|---|---|---|---|---|
| SKIP-GRAM | 0.462 / 0.520 | 0.649 / 0.123 | 0.183 / 0.268 | 0.171 / 0.847 | 0.564 / 0.286 |
| POI2VEC | 0.460 / 0.482 | 0.564 / 0.112 | 0.181 / 0.263 | 0.224 / 0.812 | 0.530 / 0.249 |
| GEO-TEASER | 0.460 / 0.470 | 0.448 / 0.116 | 0.185 / 0.266 | 0.219 / 0.818 | 0.511 / 0.194 |
| TALE | 0.461 / 0.464 | 0.375 / 0.102 | 0.183 / 0.248 | 0.233 / 0.801 | 0.504 / 0.189 |
| HIER | 0.473 / 0.547 | 0.660 / 0.119 | 0.183 / 0.291 | 0.182 / 0.859 | 0.529 / 0.229 |
| CTLE | 0.463 / 0.511 | 0.115 / 0.098 | 0.179 / 0.249 | 0.192 / 0.852 | 0.488 / 0.244 |
| DEEPMOVE | 0.460 / 0.484 | 0.370 / 0.110 | 0.183 / 0.253 | 0.249 / 0.779 | 0.503 / 0.224 |
| STAN | 0.464 / 0.509 | 0.220 / 0.099 | 0.183 / 0.250 | 0.189 / 0.854 | 0.497 / 0.248 |
| GRAPH-FLASHBACK | 0.463 / 0.506 | 0.233 / 0.099 | 0.183 / 0.251 | 0.189 / 0.853 | 0.496 / 0.248 |
| GETNEXT | 0.431 / 0.500 | 0.200 / 0.103 | 0.185 / 0.252 | 0.291 / 0.717 | 0.410 / 0.220 |
| TRAJGPT | 0.483 / 0.491 | 0.215 / 0.101 | 0.181 / 0.249 | 0.196 / 0.847 | 0.475 / 0.237 |
| **ME-POIs** (w/o $L_{\text{text-align}}$) | 0.540 / 0.703 | 0.757 / 0.154 | 0.263 / 0.337 | 0.159 / 0.878 | 0.600 / 0.308 |
| **ME-POIs** | **0.554 / 0.722** | **0.766 / 0.161** | **0.291 / 0.355** | **0.154 / 0.884** | **0.609 / 0.322** |

*Table 4.* Comparison to mobility-based models in Houston. Results averaged over 5 runs. **Best** and second best values are highlighted.

| Method | Open Hours ($\uparrow$) F1 / ($\uparrow$) AUROC | Visit Intent ($\uparrow$) F1 / ($\uparrow$) AUPRC | Busyness ($\downarrow$) MAE / ($\uparrow$) Cosine | Price Level ($\uparrow$) Accuracy / ($\uparrow$) F1 |
|---|---|---|---|---|
| SKIP-GRAM | 0.483 / 0.474 | 0.214 / 0.300 | 0.191 / 0.854 | 0.543 / 0.230 |
| POI2VEC | 0.486 / 0.503 | 0.184 / 0.298 | 0.255 / 0.778 | 0.555 / 0.270 |
| GEO-TEASER | 0.483 / 0.433 | 0.158 / 0.254 | 0.255 / 0.778 | 0.514 / 0.180 |
| TALE | 0.482 / 0.465 | 0.159 / 0.256 | 0.254 / 0.779 | 0.529 / 0.201 |
| HIER | 0.498 / 0.542 | 0.159 / 0.264 | 0.234 / 0.804 | 0.551 / 0.184 |
| CTLE | 0.306 / 0.496 | 0.183 / 0.258 | 0.195 / 0.854 | 0.511 / 0.230 |
| DEEPMOVE | 0.482 / 0.454 | 0.159 / 0.262 | 0.249 / 0.785 | 0.536 / 0.230 |
| STAN | 0.484 / 0.496 | 0.183 / 0.257 | 0.185 / 0.864 | 0.513 / 0.231 |
| GRAPH-FLASHBACK | 0.484 / 0.496 | 0.185 / 0.259 | 0.185 / 0.864 | 0.510 / 0.229 |
| GETNEXT | 0.493 / 0.551 | 0.161 / 0.293 | 0.192 / 0.857 | 0.549 / 0.180 |
| TRAJGPT | 0.483 / 0.491 | 0.179 / 0.253 | 0.188 / 0.861 | 0.534 / 0.239 |
| **ME-POIs** (w/o $L_{\text{text-align}}$) | 0.519 / 0.604 | 0.270 / 0.314 | 0.182 / 0.867 | 0.564 / 0.276 |
| **ME-POIs** | **0.582 / 0.657** | **0.306 / 0.352** | **0.177 / 0.871** | **0.590 / 0.294** |

supervises anchor POI embeddings to reflect their empirical visitation patterns, and propagates temporal signals to sparsely visited locations via its distribution transfer mechanism. Together, these components, enable ME-POIs to learn effective, POI-centric representations that outperform embeddings optimized solely for trajectory modeling.

## 4.3. Ablation Studies

**Impact of each loss term.** Table 5 reports the impact of each component on the weekly opening hours task. Starting with our main contrastive learning optimization objective, we observe that ME-POIs w/ $\mathcal{L}_{\text{ME-POI}}$, achieves strong performance that even surpasses all standard mobility-based baselines. This result, further highlights the effectiveness of our contrastive learning component. By aligning the POI prototype with all individual visit embeddings, ME-POIs aggregate diverse visitation patterns across users and timestamps into a stable, place-centric representation, that can be used to address POI-centric tasks, something that conventional sequence-based mobility models are unable to achieve. Adding the sparsity regularization term ($\mathcal{L}_{\text{KL-sparse}}$) further improves performance by stabilizing representations for

*Table 5.* Ablation of each loss term for open hours in Houston. Results averaged over 5 runs. **Best** values are highlighted.

| Method | Los Angeles ($\uparrow$) F1 / ($\uparrow$)AUROC | Houston ($\uparrow$) F1 / ($\uparrow$) AUROC |
|---|---|---|
| ME-POIs + w/ $L_{\text{ME-POI}}$ | 0.490 / 0.608 | 0.510 / 0.595 |
| + w/ $L_{\text{sparse}}$ | 0.535 / 0.701 | 0.518 / 0.603 |
| + w/ $L_{\text{anchor}}$ | 0.540 / 0.703 | 0.519 / 0.604 |
| + w/ $L_{\text{text-align}}$ | **0.554 / 0.722** | **0.582 / 0.657** |

long-tail POIs using anchor-derived visitation priors, particularly in the Los Angeles dataset where anchor coverage is denser. Incorporating the anchor alignment loss ($\mathcal{L}_{\text{KL-anchor}}$) provides additional but moderate gains, which we expect given that anchors cover only a small subset of the POIs. Finally, the text alignment loss ($\mathcal{L}_{\text{text-align}}$) further improves results by adding semantic context to our embeddings, here by aligning with OPENAI-LARGE text embeddings. Overall, each objective contributes complementary benefits, and their combination yields the best performance.

**Impact of the distribution transfer mechanism.** We evaluate the contribution of the proposed visit distribution-aware objectives by comparing a base ME-POIs model trained with only the primary contrastive loss $\mathcal{L}_{\text{ME-POI}}$ to a variant

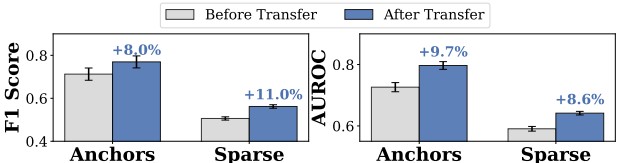

*Figure 3.* Impact of $\mathcal{L}_{\text{KL-anchor}}$ and $\mathcal{L}_{\text{KL-sparse}}$ on sparse and anchor POIs on open hours in Houston.

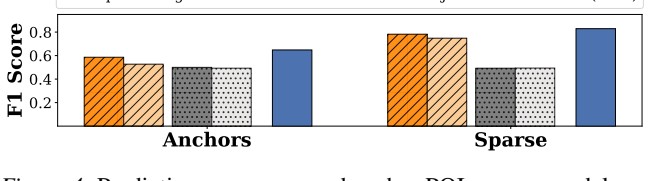

*Figure 4.* Predictions on sparse and anchor POIs across models on open hours in Houston.

that also includes the anchor and sparse distribution losses, $\mathcal{L}_{\text{KL-anchor}}$ and $\mathcal{L}_{\text{KL-sparse}}$. Results are reported separately for anchor and sparse POIs on the Houston opening hours task. As shown in Figure 3, adding the distribution transfer module consistently improves F1 and AUROC for both groups, demonstrating that multi-scale distribution transfer for sparse POIs and direct visit distribution supervision for anchor POIs improve the learned representations.

**Comparison across POI density regimes.** We further compare our ME-POIs model against selected text- (GEMINI, OPENAI-LARGE) and mobility-based (CTLE, TRAJGPT) baselines on the Houston opening hours task, reporting results separately for anchor and sparse POIs. As shown in Figure 4, ME-POIs achieves the highest F1 scores for both groups. Text-based models perform consistently across anchor and sparse POIs, as their embeddings are not affected by sparse visit data, whereas mobility-based models underperform on low-visit POIs. In contrast, ME-POIs maintains strong performance in both data regimes, highlighting the effectiveness of the multi-scale distribution transfer module in improving POI embeddings under limited supervision.

## 5. Related Work

**Static POI Representation Learning.** Existing approaches to POI representation learning primarily rely on static attributes to encode the semantic and geographic relationships between places. Several methods focus on representing location and neighborhood structure using features like geographic coordinates, geometry, proximity to other places, and local connectivity (Yan et al., 2017; Mai et al., 2020; Rußwurm et al., 2023; Klemmer et al., 2023; Siampou et al., 2025b; Chu & Shahabi, 2025). To further enrich POI representations, recent work incorporates additional context by integrating text semantics. Recent advances include (i) geospatial language models (Li et al., 2022; 2023; Yan & Lee, 2024) pretrained to improve language model performance on specialized spatial tasks, such as toponym recognition and geo-entity typing, by jointly encoding text and geographic information and (ii) approaches that extract geospatial knowledge directly from LLMs (Chen et al., 2023; Liu et al., 2024; Cheng et al., 2025). For example, GeoLLM (Manvi et al., 2024) designs spatially informed prompts to query LLMs for predicting region-specific prop-

erties (e.g., population, wealth, education) directly from LLM outputs. These methods do not incorporate dynamic human mobility patterns, which provide complementary behavioral signals and can further enhance POI embeddings.

**Mobility-Informed POI Representation Learning.** Human mobility data are widely used to model transitions between POIs. Early approaches, such as POI2Vec (Feng et al., 2017), learn co-occurrence-based embeddings from sequences of visits, while later methods incorporate spatio-temporal orderings (Zhao et al., 2017; Wan et al., 2021) or hierarchical POI structures (Shimizu et al., 2020) to improve representation granularity. CTLE (Lin et al., 2021) adopts a masked sequence modeling objective, where POI IDs and visit times are randomly masked and predicted to learn POI representations. More recently, next-location prediction models learn POI embeddings through sequence-conditioned objectives that predict the next visited POI (Feng et al., 2018; Xue et al., 2021; Rao et al., 2022; Hsu et al., 2024), sometimes augmented with graph-based structural information (Luo et al., 2021; Yang et al., 2022; Xu et al., 2024). While effective for modeling user trajectories, these objectives are trajectory-centric and thus their embeddings capture toward local transition patterns, rather than the intristic semantics of POIs, including their identity and function. ME-POIs departs from this paradigm by explicitly learning POI-centric representations.

## 6. Conclusion

We introduce ME-POIs, a pretraining framework that enriches static POI embeddings derived from text models with mobility-derived signals from visit sequences, capturing both identity and function of POIs. Our experiments show that augmenting strong text-based embeddings with ME-POIs consistently improves performance across diverse tasks, demonstrating that mobility-informed representations provide complementary information and enable a richer understanding of how places are used, beyond static metadata. These results confirm that modeling POI function is essential for generalizable and accurate POI representations. Future work will extend ME-POIs to other geospatial objects, including road segments, administrative boundaries and regions, highlighting the broader applicability of mobility-informed representation learning.

## Impact Statement

This paper presents work whose goal is to advance the field of Machine Learning. There are many potential societal consequences of our work, none of which we feel must be specifically highlighted here. Our improved, mobility-enriched POI representations could enable better location-based services and urban planning tools. Our experiments rely on aggregated, anonymized mobility data, minimizing potential privacy concerns.

## Acknowledgments

This research has been funded in part by NSF award CNS-2125530. Any opinions, findings, conclusions, or recommendations expressed in this material are those of the authors and do not necessarily reflect the views of the sponsors.

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

# A. Appendix

## A.1. Additional Details on Experimental Setup

### A.1.1. DATASET STATISTICS

Table 6 summarizes key statistics of the two mobility datasets used in our experiments. Both datasets consist of anonymized raw GPS trajectories, containing timestamped geographic coordinates and randomized device identifiers. We convert these trajectories into sequences of POI visits through a two-step preprocessing pipeline: staypoint detection and POI attribution.

For staypoint detection, we use the `trackintel` library, which implements the standard distance–time threshold method proposed by Li et al. (2008). A staypoint is identified when a user remains within a radius of 100 m for at least 5 minutes. For POI attribution, we use the POI geometries provided from SafeGraph and assign each staypoint to a POI if its location falls within the POI polygon, or otherwise to the nearest POI centroid within 100 m. Staypoints that cannot be matched to any POI are labeled as UNKNOWN visits. These visits are retained in the visit sequences to preserve temporal continuity, but are excluded from the loss computation since they lack reliable POI labels. After preprocessing, we discard visit sequences with fewer than 5 visits to ensure sufficient temporal context. POIs with at least $M$ total visits are designated as anchor POIs, while the remaining POIs are considered sparse. We set $M=100$ for Los Angeles and $M=50$ for Houston. The resulting dataset statistics, including the proportion of anchor POIs, are reported in Table 6.

The number of POIs in Los Angeles and Houston is comparable, although Los Angeles covers a larger geographic region and therefore contains more POIs. Due to its year-long temporal coverage, the Los Angeles dataset contains approximately an order of magnitude more visits than the Houston dataset, which spans 20 days.

*Table 6.* Summary of Datasets Statistics.

| Region | Time Period | Bounding Box | # POIs | # Visits | % Anchor POIs |
|---|---|---|---|---|---|
| Los Angeles | 01/01 - 12/31 2019 | [32.81, -118.94, 34.82, -117.65] | 39,557 | 6,908,365 | 9.07% |
| Houston | 03/05 - 03/26 2020 | [29.55, -95.56, 29.95, -95.16] | 28,419 | 715,604 | 7.04% |

### A.1.2. DOWNSTREAM TASK DETAILS

We evaluate our five newly introduced map enrichment tasks, chosen to comprehensively assess the quality of our POI embeddings: (i) weekly opening hours, (ii) permanent closure detection, (iii) visit intent classification, (iv) busyness estimation, and (v) price level classification. These tasks are strategically selected to capture both the functional characteristics and identity of POIs. Functional tasks, including weekly opening hours, busyness, and visit intent, reflect temporal usage patterns and user interest, while identity-focused tasks, including permanent closure and price level, capture intrinsic attributes. Labels are sourced from SafeGraph for opening hours and closure status, and from Google Maps for the remaining tasks, providing a mix of publicly available and private data. Below, we provide detailed descriptions of each of the five tasks:

- **Weekly Opening Hours.** The goal is to predict the operational schedule of each POI over a week. We represent this as a 168-dimensional binary vector, where each dimension corresponds to one hour of the week, and the value indicates whether the POI is open or closed during that hour. Ground-truth labels are derived from SafeGraph. This task evaluates the model's ability to capture temporal activity patterns of POIs. In Los Angeles, $16,692$ POIs have opening hours labels, while in Houston $14,465$ POIs have labels.

- **Permanent Closure Detection.** This is a binary classification task where the goal is to predict whether a POI is permanently closed. POIs with missing closure labels are assumed to be open. In Los Angeles, $3,807$ POIs are labeled as permanently closed. The Houston dataset does not include this task due to insufficiently reliable closure labels. This task tests whether the model can recognize POIs that do not longer exist.

- **Visit Intent Classification.** We define visit intent as a proxy for user interest in visiting a POI, measured by the average number of Google Maps direction queries to the location. Since we do not observe actual visits, this provides an indirect signal of interest. We discretize the continuous query values into four ordinal classes from low to high intent. This task evaluates the model's ability to predict POIs that attract interest from users. In Los Angeles, $22,369$ POIs have visit intent labels, and in Houston $15,632$ POIs have labels.

- **Busyness Estimation.** This task measures typical foot traffic or occupancy of a POI throughout the week, based on Google Maps' reported busyness data. For each POI, we compute an average weekly busyness signal, either as a continuous value or discretized into multiple levels for evaluation. This task assesses how well the model can capture patterns of user activity around POIs. We have 6,034 labels in Los Angeles and 5,684 in Houston.

- **Price Level Classification.** The goal is to predict the relative expense of visiting a POI, as reported by Google Maps. We map price levels into four ordinal classes ranging from low to high. This task evaluates whether the model can infer socioeconomic or commercial attributes of a location from mobility and semantic embeddings. There are 5,091 labeled POIs in Los Angeles and 4,105 in Houston.

Per-label statistics for visit intent and price level are summarized in Table 7.

*Table 7.* Visit Intent and Price Level Counts

| | Los Angeles | | Houston | |
|---|---|---|---|---|
| Class | Visit Intent | Price Level | Visit Intent | Price Level |
| 0 | 12840 | 2563 | 7158 | 2270 |
| 1 | 1376 | 2311 | 979 | 1675 |
| 2 | 5654 | 181 | 4841 | 133 |
| 3 | 2499 | 36 | 2654 | 27 |

### A.1.3. DETAILS ON BASELINES

We evaluate ME-POIs against both text- and mobility-based baselines. For text-based comparisons, we select competitive and widely adopted embedding models, including recent academic approaches, including MPNET (`all-mpnet-base-v2`), E5 (`E5-large-v2`), and GTR-T5 (`gtr-t5-large`), as well as industry-grade commercial models, including NOMIC (`nomic-embed-text-v1`), OPENAI (`text-embedding-3-small/large`), and GEMINI (`models/embedding-001`). Each model is provided with POI descriptions to generate embeddings, which are subsequently used for downstream evaluation.

We further include the following mobility baselines:

- SKIP-GRAM (Mikolov et al., 2013): Learns POI embeddings by predicting surrounding POIs in check-in sequences, capturing sequential context for mobility modeling.

- POI2VEC (Feng et al., 2017): Jointly captures user preferences, POI sequential transitions, and geographical influence to predict future visitors to a POI.

- GEO-TEASER (Zhao et al., 2017): Proposes a geo-temporal POI embedding model that captures sequential check-in contexts, day-specific temporal patterns, and geographical influence to improve POI recommendation.

- TALE (Wan et al., 2021): Learns time-aware location embeddings using a hierarchical temporal tree to improve downstream tasks such as classification, flow, and next-location prediction.

- HIER (Shimizu et al., 2020): Generates hierarchy-enhanced POI category representations by leveraging disentangled mobility sequences.

- CTLE (Lin et al., 2021): Learns POI representations via a masked language objective that predicts the location and arrival time of visits.

- DEEPMOVE (Feng et al., 2018): Uses GRU-based attention to capture both long-term periodicity and short-term sequential patterns of user trajectories.

- STAN (Luo et al., 2021): Leverages relative spatio-temporal relationships between POIs in a trajectory to improve next-location prediction.

- GRAPH-FLASHBACK (Rao et al., 2022): Enriches POI representations with a POI transition graph and combines it with sequential modeling to improve next-location recommendation.

- GETNEXT (Yang et al., 2022): Employs a graph-enhanced transformer to model global transitions, user preferences,

spatio-temporal context, and time-aware category embeddings for next-location prediction.

- TRAJGPT (Hsu et al., 2024): A transformer-based, multi-task spatiotemporal generative model that improves predictions of arrival time and duration of a user's next stay via Gaussian mixture models.

The POI representation matrix learned by each model is extracted and used for downstream evaluation.

### A.1.4. IMPLEMENTATION DETAILS & HYPERPARAMETER CONFIGURATION

**Input normalization.** All coordinates are normalized to $[0, 1]$ using the bounding box of each area of interest. We use the Space2Vec location encoder with $\lambda_{\max} = 1.4142$ (corresponding to the normalized diagonal distance), $\lambda_{\min} = 0.1$, and $64$ frequency scales. Temporal features are normalized to $[0, 1]$ by extracting the hour of day and day of week, which are encoded separately and then concatenated into a single temporal representation. For the spatial Gaussian kernels use bandwidths of 0.3, 1.0, and 3.0 km, which are normalized to align with the coordinate scale.

**Model configuration.** We set the sequence window size to $w=32$, the hidden embedding dimension to $d_h=512$, and the text embedding dimension to $d_u=768$. The Transformer encoder backbone consists of $N=4$ layers with $i=8$ attention heads and a feedforward hidden dimension of 1024. All MLP modules use a single hidden layer with 256 units and ReLU activation. Overall, the ME-POIs framework is lightweight with $\sim 53.7$ M parameters, well within standard computational budgets.

**Training details.** The model is pretrained on the full visit sequence dataset and subsequently fine-tuned using a 60/20/20 train/validation/test split. We use the Adafactor optimizer during pretraining with a learning rate of $1e{-}3$, and AdamW during fine-tuning with a learning rate of $1e{-}5$. Model is pretrained for 20 epochs, and fine-tuned for 100 epochs with early stopping. Unless otherwise stated, we set $\lambda_\alpha = \lambda_s = \lambda_t = 1$.

### A.1.5. EXPERIMENTAL ENVIRONMENT

We implement our models in PyTorch 2.6.0 on a Debian Linux server, equipped with 50 GB RAM, 8 vCPUs (Intel Xeon @ 2.30 GHz), and an NVIDIA Tesla V100–SXM2–16GB GPU (CUDA 13.0).

### A.2. Prompt Examples for Text Embedding Models

We construct text prompts for each POI following the GeoLLM (Manvi et al., 2024) approach, which incorporates both (i) POI information, including coordinates, category, and address, which we obtain from Safegraph and (ii) neighborhood context, including the name, distance, and direction of the 10 closest POIs. This prompt design has been shown to effectively extracts geospatial knowledge, producing text embeddings that captures rich semantic and contextual information. We then query text embedding models (e.g., OPENAI and GEMINI), and set the output dimension to 768, to ensure a fair comparison across models. An example prompt for the TACO MAN POI in Los Angeles is illustrated in Figure 5.

> **Taco Man** (Restaurants and Other Eating Places). Coordinates: 34.062307, -118.197612.
> Address: 1602 N Soto St, Los Angeles, CA, 90033.
>
> **Nearby Places:**
> 0.0 km West: Tacos La Guera;
> 0.0 km West-Southwest: Soto Liquor Market;
> 0.1 km West: DaVita;
> 0.1 km West: Davita Trc Usc Kidney Center;
> 0.2 km North-Northeast: Ai Food Corporation;
> 0.2 km West: USC Occupational Therapy Faculty Practice;
> 0.2 km West: Molecular Imaging Center;
> 0.2 km West-Southwest: Bright Horizons Usc Hsc Infant Care Center;
> 0.2 km West-Southwest: Bright Horizons Usc Hsc Child Development Ctr;
> 0.3 km Northeast: Cardinal Moving Systems.

*Figure 5.* Example prompt for Taco Man POI in Los Angeles.

### A.3. Computational Efficiency

The pre-training cost of ME-POIs is dominated by running the visit encoder on sequences of visits. For a sequence length of $L$ and an embedding dimension $d$, the overall computation complexity is $O(L^2 \cdot d + L \cdot d^2)$ for. The contrastive module

operates only over in-batch negatives: for a batch of $B$ visits containing $U$ unique POIs, its cost is $O(B \cdot U \cdot d)$, which in practice remains lightweight and independent of the full POI set size. Note that the # of unique POIs in the batch is less than or equal to # of visits in the batch. The POI anchor distributions and multiscale kernels are precomputed only once offline, with computation complexity $O(M \cdot |\mathcal{P}_{\text{anchor}}| \cdot |\mathcal{P}_{\text{sparse}}|)$ for $M$ scales.

### A.4. Effect of Training Data Duration

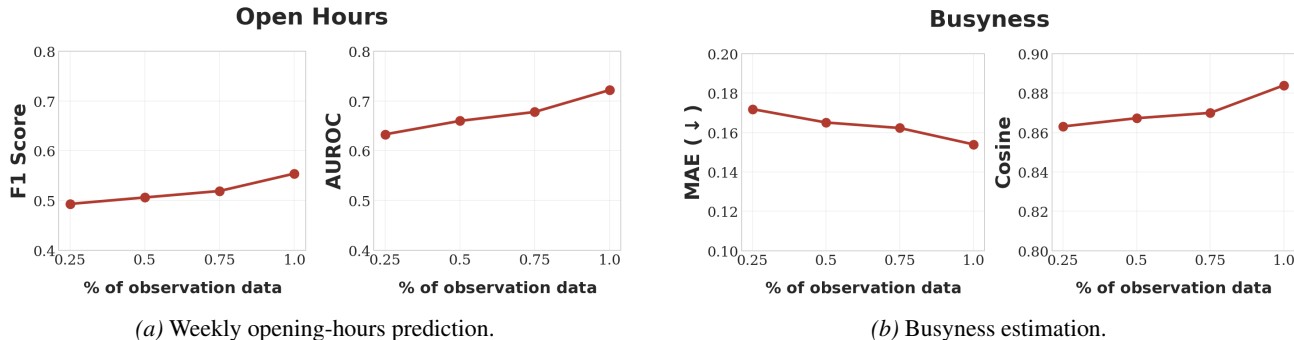

*(a)* Weekly opening-hours prediction.        *(b)* Busyness estimation.

*Figure 6.* Effect of mobility observation duration on ME-POI representation quality on the Los Angeles dataset. We pretrain ME-POIs using 25%, 50%, and 75% of the 12-month observation period and evaluate the learned representations on weekly opening-hours prediction and busyness estimation. Longer observation windows generally improve performance, while ME-POIs remains effective even with only 25% of the data.

We further study how the duration of mobility observations affects the quality of the learned POI representations. On the Los Angeles dataset, we pretrain ME-POIs using only 25%, 50%, and 75% of the 12-month observation period, and evaluate the resulting representations on weekly opening-hours prediction and busyness estimation. Figure 6 shows that representation quality generally improves as the observation window increases. This suggests that longer mobility histories provide richer signals, enabling the model to better capture longitudinal and seasonal usage patterns while reducing the effect of transient noise. Notably, ME-POIs remains effective even when pretrained with only 25% of the data, corresponding to approximately three months of observations. This indicates that a relatively short mobility window already contains sufficient context for learning useful POI representations. Moreover, ME-POIs pretrained on only 25% of the data still outperforms mobility baselines pretrained on the full 12-month dataset. We attribute this to the difference in learning objectives. Standard mobility models are typically optimized for trajectory or next-location prediction, which emphasizes local transition patterns and transient sequential context. In contrast, ME-POIs explicitly learns context-independent, POI-centric representations by aggregating visit-level signals into a global POI prototype. This design allows the model to capture the POI function and identity, even under reduced training data.

