# OpenReview forum: "Mobility-Embedded POIs: Learning What A Place Is and How It Is Used from Human Movement"
_ICML.cc/2026/Conference — ICML 2026 regular_

### Official Review · Reviewer_xvcK · 2026-03-06

**Soundness:** 4
**Presentation:** 4
**Significance:** 4
**Originality:** 4
**Overall Recommendation:** 5
**Confidence:** 4

**Summary:**

This paper proposes an embedding framework for points-of-interest which incorporate information and their static identity and their dynamic function. Their framework, Mobility-Embedded POIs, is trained using a set of static POI embeddings obtained from a pre-training text embedding model, and a collection of of user visit sequences, which are each make up of events detailing the arrival and departure times of the user to a POI.

A vector representation of each visit is obtained by concatenating representations of the geographic coordinates (obtained via the Space2Vec encoder) and of the arrival and departure times (obtained via the Time2Vec encoder). Positional encodings are added to each representation which are head into a transformer to obtain contextualised visit embeddings. A single representation for POI is obtained from the contextualised visit embeddings via contrastive learning. The authors propose a multi-scale distribution transfer mechanism for sparsely visited POIs and direct supervision mechanism to visitation distribution of frequently visited POIs to the visitation distributions implied by their embeddings. Finally the propose a mechanism to align these mobility embeddings with the static POI embeddings obtained via a pre-trained text embedding model (if I've understood correctly, $z_p^{\text{static}}$ and $z_p^{\text{text}}$ are the same objects?)

They design 5 regression/classification benchmarks to assess the fidelity of their embeddings, which they use to compare their method against a large suite of alterantive embedding methods. Their method significantly outperforms the alternatives.

Finally, they perform an ablation study which studies the impact of the different mechanisms in their model.

**Compliance With Llm Reviewing Policy:**

Affirmed.

**Final Justification:**

I continue to champion this paper. See my comments in the AC-reviewer discussion.

**Key Questions For Authors:**

- Just to make sure I've understood, can you clarify whether $z_p^{\text{static}}$ and $z_p^{\text{text}}$ are indeed the same objects?
- The impact of this work would be significantly increased by open-source code, a public-facing API and/or a dataset of POI embeddings. Do you can to release any of these?

**Limitations:**

There is no explicit discussion of limitations.

**Strengths And Weaknesses:**

This problem addressed by this paper, of learning Point-of-Interest embeddings which encode both the static identity, and dynamic function of a place is an important problem which is relevant to a wide-range of practitioners, and I think the authors do a really nice job of articulating their motivation and the significance and potential impact of their work. In particular, their explanation of "what a place is" and "how it is used" using clear, simple language and examples gives the reader a comprehensive picture of the problem they are trying to solve. Overall, the paper is very clear and well-written.

The model arcitecture they propose is highly original and pragmatic, and each step and its motivation and explained clearly and accurately. The downstream performance benchmark they design to evaluate their method are relevant to the tasks that these embeddings would be used for by practitioners, and I think they serve as good indicators of embedding fidelity. The authors compare against a large suite of alternative methods, and their proposes embeddings perform significantly better than these according to their metrics. I would say that these experiments are convincing.

Additionally, the ablation studied highlight the impact of each term in their model which is enlightening.

Overall, I think I think this is a novel method, a well written paper and has high potential impact, therefore I would recommend it for acceptance.

---

> ### Author Rebuttal · Authors · 2026-03-31
>
> We thank the reviewer for the encouraging comments! Please find the answers to your questions below:
>
> **Reviewer Q1:** “[...] can you clarify whether  $z_p^{static}$ and $z_p^{text}$ are indeed the same objects?” \
> **Answer:** Thank you for catching that. Yes, these are indeed the same objects. We will correct this to keep the notation consistent throughout the paper.
>
> ---
> **Reviewer Q2:** “Do you can to release POI embeddings and code?” \
> **Answer:** Yes, we plan to publicly release both the pretrained POI-embeddings and open-source our code with links in the published paper.

---

> > ### Author Rebuttal · Reviewer_xvcK · 2026-04-01
> >
> > Thanks for the response. I continue to strongly support this paper.

---

### Official Review · Reviewer_v7yi · 2026-03-11

**Soundness:** 3
**Presentation:** 3
**Significance:** 3
**Originality:** 3
**Overall Recommendation:** 4
**Confidence:** 2

**Summary:**

This paper introduces ME-POIs, a pretraining framework that learns POI-centric, context-independent embeddings capturing both place identity and place function by augmenting language-model POI embeddings with large-scale human mobility data. The key idea is to represent each visit as a temporally contextualized embedding and contrastively align these visit embeddings with a learnable global POT prototype so that the prototype aggregates stable usage patterns across users and time while remaining semantically grounded. In experiments, the proposed method consistently outperforms both text-only and mobility-only baselines on five newly implemented map-entichment tasks.

**Compliance With Llm Reviewing Policy:**

Affirmed.

**Final Justification:**

I still believe that applying the method to the standard benchmarks for POI (including next-POI task) would further strengthen the contribution, but I understand that this can be left for future work. As my main concerns have been resolved, I will maintain my positive score.

**Key Questions For Authors:**

(Related to Weakness 1) I am curious whether similar improvements would also be observed on existing benchmarks commonly used in POI or mobility representation learning.

**Limitations:**

yes

**Strengths And Weaknesses:**

Strengths

* The motivation is clear and intuitive. The paper highlights the importance of combining static semantic information (e.g., textual metadata) with behavioral signals derived from human mobility to learn richer POI representations that capture both place identity and functional usage.

* The paper provides comprehensive experimental analysis of the proposed method. In particular, the authors conduct detailed ablation studies that examine the contribution of each component (e.g., contrastive alignment, distribution transfer for sparse POIs, and text alignment), offering useful insights into the effectiveness of the design choices.

Weaknesses

* The evaluation relies exclusively on newly proposed map enrichment tasks introduced in this work. While these tasks are well motivated, it is unclear whether the proposed method would demonstrate similar advantages on established benchmarks commonly used in POI or mobility representation learning. Including experiments on existing benchmarks would strengthen the evidence that the improvements are not specific to the newly introduced tasks.

* The experiments use mobility datasets covering one year (Los Angeles) and 20 days (Houston). However, it would be useful to analyze how the performance depends on the length of the mobility data used for training. For example, understanding how accuracy changes with shorter observation windows (e.g., a few weeks or months) would provide valuable insights for practical deployment scenarios where long-term mobility data may not always be available.

---

> ### Author Rebuttal · Authors · 2026-03-31
>
> We thank the reviewer for their questions, please see our detailed responses below:
>
> **Reviewer Q1:** “I am curious whether similar improvements would also be observed on existing benchmarks commonly used in POI or mobility representation learning.” \
> **Answer:** To our knowledge, standard benchmarks for POI and mobility representation learning primarily include two tasks: POI flow/visit-pattern prediction (e.g., [1]) and next-POI prediction (e.g., [2]) . POI flow prediction is equivalent to our weekly busyness estimation task, where we show that ME-POIs outperform existing mobility-based models. As for next-POI prediction, ME-POIs is not designed for this task, since our framework aggregates multiple user visits into a single POI representation. Next-POI prediction, in contrast, requires sequence-level modeling that captures the immediate temporal dependencies between consecutive visits. Our ME-POIs, on the other hand, aim to provide a universal, context-independent encoding of POI function.
>
> ---
> **Reviewer Q2:** “[...] it would be useful to analyze how the performance depends on the length of the mobility data used for training.”\
> **Answer:** Thank you for the suggestion! We have added an ablation study on the Los Angeles dataset by subsampling the observation period into fractions of 25%, 50%, and 75%. We present the results for the tasks of weekly open hours and busyness estimation, which can be found here: [fig_1](https://anonymous.4open.science/r/rebuttal-0362/ablation_varying_obs_window/ablation_data_period_busyness.png) & [fig_2](https://anonymous.4open.science/r/rebuttal-0362/ablation_varying_obs_window/ablation_data_period_open_hours.png). Overall, we observe a positive correlation between data duration and representation quality. While ME-POIs performs sufficiently even with 25% of data (~3 months) (which indicates that it captures sufficient functional context), a longer observation window further refines these representations by capturing longitudinal and seasonal usage patterns while filtering out transient noise. Interestingly, ME-POIs pretrained on only 25% of the data still outperforms mobility baselines pretrained on the full 12-month dataset. We attribute this to the fact that standard mobility models are optimized for trajectory prediction (next-location) and thus focus on local transition patterns, which limits their ability to model the overall POI function and identity. In contrast, ME-POIs explicitly learns context-independent, POI-centric representations by aggregating visit-level signals into a global prototype. We believe these results further strengthen our model design and plan to include them into the revised paper.
>
> [1] Feng, Shanshan, et al. "Poi2vec: Geographical latent representation for predicting future visitors." AAAI 2017. \
> [2] Hsu, Shang-Ling, et al. "Trajgpt: Controlled synthetic trajectory generation using a multitask transformer-based spatiotemporal model." SIGSPATIAL 2024.

---

> > ### Author Rebuttal · Reviewer_v7yi · 2026-03-31
> >
> > Thank you for your rebuttal. I still believe that applying the method to the standard benchmarks for POI (including next-POI task) would further strengthen the contribution, but I understand that this can be left for future work. As my main concerns have been resolved, I will maintain my positive score.

---

### Official Review · Reviewer_mTu9 · 2026-03-12

**Soundness:** 2
**Presentation:** 3
**Significance:** 2
**Originality:** 2
**Overall Recommendation:** 3
**Confidence:** 5

**Summary:**

This paper introduces Mobility-Embedded POIs (ME-POIs), a novel framework for learning *behavior-aware* point-of-interest (POI) representations by embedding large-scale human mobility patterns into otherwise static POI embeddings. The core motivation is that conventional POI representations—derived from text, images, or spatial coordinates—capture *what a place is*, but fail to describe *how a place is used*. To address this gap, the authors model mobility sequences as temporal visit events and design a contrastive sequence encoding mechanism that aligns context-aware visit embeddings with global POI embeddings.

**Compliance With Llm Reviewing Policy:**

Affirmed.

**Final Justification:**

In my view, this work is more of an integration of existing methods and lacks sufficient novelty and innovation. So, I maintain my original score and recommend weak rejection.

**Key Questions For Authors:**

How sensitive is the model to the threshold used for defining anchor POIs?

**Limitations:**

Yes.

**Strengths And Weaknesses:**

Strengths:
1. The work emphasizes the joint modeling of semantic information and behavioral patterns in POI representation learning, which has been largely overlooked in prior POI representation work.
2. The spatially grounded multi-scale transfer of visit-time distributions is conceptually well-motivated and demonstrates strong empirical effectiveness.
3. The experimental evaluation is comprehensive. The authors conduct experiments across multiple settings, including text-based and mobility-based scenarios, which helps demonstrate the effectiveness and robustness of the proposed framework.
Weaknesses:
1. The loss for sparse POIs relies on prior visitation distributions transferred from anchor POIs according to spatial proximity, yet the paper does not provide sufficient theoretical analysis or empirical evidence to justify the validity of this assumption.
2. Although the overall framework is meaningful, most of the individual components are relatively standard, including the Transformer-based sequence encoder, InfoNCE contrastive learning. As a result, the novelty of the work mainly lies in the integration of these existing techniques rather than introducing fundamentally new modeling methods or frameworks.
3. The dataset of veraset is not freely available to the public. The link should at least include a demo of the data, but this paper does not provide it.

---

> ### Author Rebuttal · Authors · 2026-03-31
>
> We thank the reviewer for their questions, please see our detailed responses below:
>
> **Reviewer W1:** “The loss for sparse POIs relies on prior visitation distributions transferred from anchor POIs according to spatial proximity, yet the paper does not provide sufficient theoretical analysis or empirical evidence to justify the validity of this assumption.” \
> **Answer:** Our assumption that spatial proximity correlates with functional similarity is grounded in established geographic principles, specifically Tobler’s First and Third Law of Geography, which state that entities in similar geographic contexts tend to exhibit similar functional attributes. This concept is widely validated in urban computing (e.g., [1,2,3,4]). While we do not provide formal theoretical guarantees for this assumption, we would like to clarify that the paper does provide substantial empirical evidence supporting its validity. In particular, Sec. 4.2 shows through several ablation studies that: *(1)* the multi-scale transfer loss improves overall model performance (Table 5), *(2)* it specifically improves performance for sparsely visited POIs (Fig 3), and *(3)* the full ME-POIs model outperforms both text- and mobility-based baselines on sparse POI predictions (Fig 4). Together, these results provide empirical support that the proposed transfer mechanism stabilizes sparse POI representations and improves predictive performance.
>
> ---
> **Reviewer W2:** “Most of the individual components are relatively standard, including the Transformer-based sequence encoder, InfoNCE contrastive learning.” \
> **Answer:** We note that the use of standard building blocks such as Transformers and InfoNCE should not be taken as evidence of limited novelty, as our contribution lies in the unique combination of these blocks to solve fundamental challenges in POI representation that as we demonstrate in our experiments neither off-the-shelf text models nor prior mobility models can address. Overall, the novelties introduced in this paper are several: *(1)* We propose a new learning objective for POI representations, which produces embeddings that capture both the identity and the function of POIs from visit sequences, moving beyond traditional methods that only model local trajectory transitions. *(2)* Our contrastive learning objective is based on InfoNCE, but we adapt it to align each individual visit embedding with a single, learnable POI embedding. This allows ME-POIs to aggregate diverse, noisy user visits into a stable, global representation of each POI. To our knowledge, this formulation has not been applied to POI representation learning before. *(3)* We introduce a multi-scale visit distribution transfer mechanism to address sparsity in POI visits. *(4)* We evaluate ME-POIs  on multiple, newly introduced tasks, including over strong text-based embeddings like OpenAI and Gemini, which highlights the practical impact of our work.
>
> ---
> **Reviewer W3:** “The dataset of veraset is not freely available to the public.” \
> **Answer:** While the Veraset dataset is proprietary, it is widely used in human mobility research and is available to researchers through a standard data-use request. Furthermore, Veraset provides a free demo of the data (https://www.veraset.com/about/our-data). We will include this link of the data demo in our paper.
>
> ---
> **Reviewer Q1:** “How sensitive is the model to the threshold used for defining anchor POIs?” \
> **Answer:** Thank you for raising this point! We have added an ablation study on the Los Angeles dataset, where we varied $M \in$ {20, 50, 100, 200, 500}, where $M$ is the number of visits required for a POI to serve as an anchor. The results can be found in here: [fig_1](https://tinyurl.com/49ay4r5v) & [fig_2](https://tinyurl.com/yxbn6pn3). We observe that the downstream performance of ME-POIs remains stable across different values of $M$, with the best performance achieved at our default setting ($M=100$). This stability is expected, as our primary objective for learning the POI embeddings is the contrastive loss ($L_{ME-POI}$), while $L_{KL-sparse}$ and $L_{KL-anchor}$ serve as refinements. Overall, setting $M$ to very low values will result in more anchors, which may introduce noise (due to their few visits) into the empirical distributions $r_{p_a}$. Setting $M$ to very high values will reduce the number of available anchors, which limits the density of the spatial kernels and may result in sparse POIs receiving less relevant temporal priors from distant anchors.
>
> [1] "A computer movie simulating urban growth in the Detroit region." Economic geography. 1970\
> [2] "Poi2vec: Geographical latent representation for predicting future visitors." AAAI 2017.\
> [3] "Urban region representation learning with openstreetmap building footprints." KDD 2023.\
> [4] "Road network representation learning with the third law of geography." NeurIPS 2024.

---

> > ### Author Rebuttal · Reviewer_mTu9 · 2026-04-07
> >
> > Thank you for your reply. While I acknowledge the efforts made in this work, I still find that the proposed method is relatively straightforward and mainly consists of commonly used or somewhat outdated techniques. The solution for the long-tailed distribution problem is also conventional. In my view, this work is more like an integration of existing approaches and lacks sufficient novelty and innovation. Therefore, I have to maintain my original score.

---

### Official Review · Reviewer_QTYJ · 2026-03-13

**Soundness:** 3
**Presentation:** 3
**Significance:** 2
**Originality:** 2
**Overall Recommendation:** 4
**Confidence:** 3

**Summary:**

This paper proposes ME-POIs, a framework that augments text-based POI embeddings with large-scale human mobility data to learn POI-centric representations capturing both identity and function. Experiments on several map enrichment tasks show consistent improvements over text-only and mobility-only baselines.

**Compliance With Llm Reviewing Policy:**

Affirmed.

**Final Justification:**

I thank the authors for the helpful clarification, which improved my understanding of the role of the multi-scale relation modeling. I still believe a more unified representation mechanism could further strengthen the work, but I maintain my positive score.

**Key Questions For Authors:**

•	What is the relationship between the multi-scale POI information transfer in Section 3.3 and the Space2Vec positional encoding in Section 3.1? Since Space2Vec already encodes spatial information at multiple scales, can the model naturally capture cross-scale POI spatial patterns through the Transformer encoder?

•	Text embeddings may already encode some target attributes. For models like OpenAI or Gemini, providing the POI name and location could expose web-derived signals about price level or opening hours, which may introduce bias and make comparisons with mobility-based models less fair.

•	Did the authors test cross-city transfer, such as train on Los Angeles and test on Houston, or reverse? Without such results, the claim of “generalizable” POI representations seems insufficiently supported.

**Limitations:**

Yes

**Strengths And Weaknesses:**

**Strengths**:

•	The distinction between trajectory and POI representation learning is clear and well-motivated.

•	The multi-scale distribution transfer for sparse POIs is practical and appears effective for long-tail settings.

•	Experiments are fairly comprehensive and show improvements across multiple downstream tasks.

**Weaknesses**:

•	The paper is well organized, but its novelty seems incremental and not clearly beyond existing POI embedding frameworks.

•	The evaluation is limited to two U.S. cities, so the claim of learning general POI representations is not yet fully supported, especially without cross-city validation.

---

> ### Author Rebuttal · Authors · 2026-03-31
>
> We thank the reviewer for the thought provoking questions, please see our detailed responses below:
>
> **Reviewer W1:** "The paper is well organized, but its novelty seems incremental [...]''. \
> **Authors:** We appreciate the reviewer’s feedback, but we would like to clarify that ME-POIs introduces several fundamental shifts beyond existing POI embedding frameworks: (1) We propose a mobility-based objective to learn POI-centric embeddings that encode the identity and function of POIs, unlike existing methods that focus exclusively on capturing local trajectory transitions in their representations. (2) We introduce a contrastive learning objective to align each individual visit embedding with a single, learnable POI embedding. This allows ME-POIs to aggregate diverse, noisy user visits into a stable, global POI representation. To our knowledge, this formulation has not been applied to POI representation learning before. (3) We introduce a multi-scale visit distribution transfer mechanism to address visit sparsity. (4) We evaluate ME-POIs  on multiple, new tasks, including over strong text-based embeddings like OpenAI and Gemini, which highlights the practical impact of our work. We further show that neither off-the-shelf text nor prior mobility models can address these tasks, demonstrating that ME-POIs solves fundamental challenges in POI representation that were previously unaddressed.
>
> ---
> **Reviewer Q1:** “What is the relationship between the multi-scale POI information transfer in Section 3.3 and the Space2Vec positional encoding in Section 3.1?” \
> **Authors:** In our framework, Space2Vec (Sec 3.1) provides a feature-level location encoding of the POIs within a user’s sequence. The Transformer encoder leverages these representations through self-attention to capture spatio-temporal patterns within those sequences, such as correlations between spatial proximity and co-visitation. However, because the input is organized as individual, user-centric sequences, the model at this stage does not explicitly aggregate information across all visits to the same POI. This is a particular issue for sparsely visited POIs (which are the majority), because they appear only a few times and may never co-occur with others in the same sequence. The proposed multi-scale mechanism in Sec. 3.3 addresses this by introducing a cross-user signal that allows these sparse POIs to "borrow" stable visitation priors from nearby, data-rich anchors. Thus, while Space2Vec encodes local location information at the input level, Sec. 3.3 introduces a mechanism for neighborhood-informed knowledge transfer, which helps sparse POIs inherit stable visitation patterns from nearby anchors.
>
> ---
> **Reviewer Q2:** “For models like OpenAI or Gemini, providing the POI name and location could expose web-derived signals about price level or opening hours, which may introduce bias and make comparisons with mobility-based models less fair.” \
> **Authors:** We agree with the reviewer that large-scale text models may have indirect access to some target attributes via their training data (as mentioned in Sec. 4.2). But note that in our experiments, we do not directly compare text and mobility-based models, rather, we show that adding ME-POIs on top of text provides an extra and complementary signal that improves these text-based representations. For example, for the tasks of open hours and price level, adding ME-POIs increases F1 by up to 15.8% and 75.1%, respectively. This confirms that while text embeddings capture some attribute-related information, they are often incomplete and biased toward popular POIs, and the grounding representations in real-world visitation behavior allow our approach to fill these metadata gaps. Finally, although not intended as a direct comparison, the fact that ME-POIs without text supervision ($L_{text-align}$) can match or even outperform text-only models in some cases, highlights the unique value of incorporating mobility signals through our proposed framework even more.
>
> ---
> **Reviewer Q3:** “Did the authors test cross-city transfer, such as train on Los Angeles and test on Houston, or reverse? Without such results, the claim of “generalizable” POI representations seems insufficiently supported.” \
> **Authors:** Thank you for raising this point. By using the term “generalizable” we meant that the learned POI embeddings are broadly useful across multiple downstream tasks, rather than that they demonstrate cross-city transferability in the zero-shot sense. We agree that these are different notions, as support for multiple downstream tasks speaks to the generality/utility of the representation, while cross-city training/testing evaluates its transferability. We will revise the paper to make this distinction explicit and avoid ambiguity. We note that cross-city transfer is a substantially more challenging problem due to shifts in urban structure and mobility behavior, and we view it as an important direction for future work.

---

> > ### Author Rebuttal · Reviewer_QTYJ · 2026-04-04
> >
> > Thanks for your response. The comment I provide here is more of a suggestion for clarification rather than a required issue that must be fully resolved.
> >
> > Multi-scale analysis is indeed important here. I can understand the motivation of Sec. 3.3 for sparse POIs, but I do not feel the current response fully addresses the question. At the moment, this component reads somewhat like an additional patch. Since Sec. 3.1 already introduces multi-scale spatial encoding, one natural question is whether the main model should already be able to learn such multi-scale spatial patterns internally. If not, it would be helpful to explain more clearly what is fundamentally missing from that modeling pipeline, and why the extra transfer mechanism in Sec. 3.3 is necessary.
> >
> > For the generalizability part, I understand that the goal of the embedding is to support downstream tasks. I am just curious about how such information could transfer across different areas or cities. Intuitively, geographic locality may limit the representation, so this type of POI embedding may remain most useful in data-rich regions where it is trained, while being harder to extend to data-sparse areas. In that sense, I appreciate the authors’ clarification that “generalizable” here refers to usefulness across tasks rather than cross-city transferability, and I think making this distinction explicit in the paper would be helpful.
> >
> > Thanks again for the authors’ efforts. I should also note that I am not deeply familiar with this specific subfield, so I will maintain my current score.

---

> > > ### Author Response · Authors · 2026-04-05
> > >
> > > We thank the reviewer for the positive feedback!
> > >
> > > We would like to provide clarification regarding the *multi-scale analysis*:
> > > The key distinction is between the **user-centric organization of the input** and the **POI-centric representation that the model ultimately aims to learn.** In Section 3.1, Space2Vec is applied to each POI occurrence within a user’s visit sequence, so it provides a positional encoding that helps the Transformer capture spatial relations **within that sequence**, e.g., whether nearby POIs tend to be visited in related temporal contexts. However, the Transformer input remains user-centric: each training instance is one user trajectory, not a collection of all visits associated with the same POI.
> > > By contrast, the objective of our framework is to learn **POI embeddings**, i.e., representations that reflect how each POI participates in mobility patterns **across all users**. This creates a gap: even if Space2Vec enriches each POI occurrence locally, it does not explicitly aggregate evidence from multiple users who visited the same POI. This is especially problematic for sparse POIs, which may appear only a few times and may never occur often enough in user sequences to learn a stable embedding. The multi-scale POI information transfer in Section 3.3 is introduced precisely to address this issue: it operates at the POI level and allows sparse POIs to borrow stable visitation priors from nearby, data-rich POIs.
> > > For example, consider the user-centric input sequences: \
> > > user-1: POI1, POI2, POI3, POI4 \
> > > user-2: POI6, POI2, POI1 \
> > > user-3: POI3, POI6, POI4 \
> > > These are the units processed by the Transformer. However, the desired output is a POI-centric representation, such as an embedding for POI1 informed by visits from user-1 and user-2, or for POI3 informed by visits from user-1 and user-3. Space2Vec helps encode where POI1 or POI3 is located when it appears in a sequence, but it does not by itself provide an explicit mechanism to consolidate POI-level evidence across users. Section 3.3 provides that missing mechanism through neighborhood-based transfer.
> > >
> > > In short, **Space2Vec encodes spatial context at the input level within user trajectories, whereas the multi-scale transfer module operates at the POI level to improve the learned POI embeddings through cross-user, neighborhood-informed knowledge transfer.**
> > >
> > > Lastly, regarding the *transferability* of the embeddings, we agree with the reviewer's intuition that the geographic locality of mobility patterns can inherently limit off-the-shelf cross-city transfer. For that reason, we do not expect the POI representations to be transferable to different cities without explicit alignment. As suggested, we will explicitly clarify in the paper that *our goal is to produce representations that can be utilized across diverse downstream tasks*.

---

### Decision · Program_Chairs · 2026-04-30

**Decision:**

Accept (regular)

**Comment:**

After the discussion phase, the majority of reviewers recommended acceptance (Accept, 2x Weak Accept, Weak Reject), finding the task to be interesting, the paper well written, the proposed approach well motivated with novelty, and the experimental analysis to be comprehensive. The dissenting reviewer, however, found there to be a lack of theoretical analysis for the proposed objective function, limited architectural novelty, and had concerns about the use of a proprietary dataset. The rebuttal addressed many of the reviewer concerns (e.g., by adding a new experiment studying the impact of the amount of historical mobility data available during training); however, the reviews remained divergent. A discussion examined the concerns of the dissenting reviewer as well as the corresponding author response, which addressed each concern (e.g., by summarizing contributions, mentioning that the dataset used is available for research use by request). Ultimately, the affirming reviewers agreed on some points but still argued for acceptance. As a result, the AC decided to accept the paper. Please take the reviewer feedback into account when preparing the camera-ready version.